# Validation of a Primary Production Algorithm of Vertically Generalized Production Model Derived from Multi-Satellite Data around the Waters of Taiwan

**Kuo-Wei Lan** [1,2](), **Li-Jhih Lian** [3], **Chun-Huei Li** [4,*](), **Po-Yuan Hsiao** [1]() **and Sha-Yan Cheng** [1]

[1] Department of Environmental Biology Fisheries Science, National Taiwan Ocean University, 2 Pei-Ning Rd., Keelung 20224, Taiwan; kwlan@mail.ntou.edu.tw (K.-W.L.); 10831005@mail.ntou.edu.tw (P.-Y.H.); eric@mail.ntou.edu.tw (S.-Y.C.)

[2] Center of Excellence for Oceans, National Taiwan Ocean University, 2 Pei-Ning Rd., Keelung 20224, Taiwan

[3] Taiwan Cross-Strait Fisheries Cooperation and Development Foundation, 100 Heping W. Rd, Taipei 10070, Taiwan; juno516874@gmail.com

[4] Marine Fisheries Division, Fisheries Research Institute, Council of Agriculture, 199 Hou-Ih Rd, Keelung 20246, Taiwan

\* Correspondence: chli@mail.tfrin.gov.tw; Tel.: +886-2-24622101 (ext. 2304)

**Abstract:** Basin-scale sampling for high frequency oceanic primary production (PP) is available from satellites and must achieve a strong match-up with in situ observations. This study evaluated a regionally high-resolution satellite-derived PP using a vertically generalized production model (VGPM) with in situ PP. The aim was to compare the root mean square difference (RMSD) and relative percent bias (Bias) in different water masses around Taiwan. Determined using light–dark bottle methods, the spatial distribution of VGPM derived from different Chl-a data of MODIS Aqua ($PP_A$), MODIS Terra ($PP_T$), and averaged MODIS Aqua and Terra ($PP_{A\&T}$) exhibited similar seasonal patterns with in situ PP. The three types of satellite-derived PPs were linearly correlated with in situ PPs, the coefficients of which were higher throughout the year in $PP_{A\&T}$ ($r^2 = 0.61$) than in $PP_A$ ($r^2 = 0.42$) and $PP_T$ ($r^2 = 0.38$), respectively. The seasonal RMSR and bias for the satellite-derived PPs were in the range of 0.03 to 0.09 and −0.14 to −0.39, respectively, which suggests the $PP_{A\&T}$ produces slightly more accurate PP measurements than $PP_A$ and $PP_T$. On the basis of environmental conditions, the subareas were further divided into China Coast water, Taiwan Strait water, Northeastern upwelling water, and Kuroshio water. The VPGM PP in the four subareas displayed similar features to Chl-a variations, with the highest PP in the China Coast water and lowest PP in the Kuroshio water. The RMSD was higher in the Kuroshio water with an almost negative bias. The $PP_A$ exhibited significant correlations with in situ PP in the subareas; however, the sampling locations were insufficient to yield significant results in the China Coast water.

**Keywords:** primary productivity; vertically generalized production model; waters around Taiwan; MODIS Aqua and Terra

## 1. Introduction

Primary production (PP) refers to the production of organic carbon during photosynthesis [1]. It sets the upper limit for ocean productivity and is an essential measure of the ocean's capacity to transform carbon dioxide into particulate organic carbon at the base of the food web [2–4]. From a bottom-up perspective [2,5,6], PP is also a good predictor of the potential yield of the world's oceans.

In the marine environment, in situ measurements of PP are taken using materials such as $^{14}$C [7], $^{13}$C [8], chlorophyll a (Chl-a) fluorescence [9], and oxygen isotopes [10]. These shipboard measurements of the snapshot sections vary over short temporal and spatial scales [7–10]. Furthermore, it can be time-consuming to represent minute fractions of ecosystems [6,11]. Scaling these relatively separate in situ measurements of the snapshot sections to a regional scale, let alone basin or global scale projections, therefore remains a significant challenge and needs to rely on remote sensing data and models [11–14].

Basin-scale sampling for high-frequency PP is available from satellites [13]. Ocean color images derived from remote sensing are ideal for assessing PP on a regional to global scale, and provide high-quality spatial and temporal coverage that give daily estimations of the attenuation coefficient, phytoplankton biomass, and photosynthetically available radiation (PAR) [14]. Remote sensing of ocean color cannot provide adequate information on oceanic PP without the support of models and sea truth data [15,16]. Several analytical, empirical, and bio-optical models are currently used to determine ocean PP [2,15,17,18]. Chl-a based models were chosen for this study; a large archive of regional pigment data is available for use [19,20] and only a limited amount of bio-optical data is. The vertically generalized production model (VGPM) formulated by Behrenfeld and Falkowski [19] is among the most commonly used and simplest models for estimating PP from Chl-a data obtained from satellites.

The VGPM is a vertically integrated and light-dependent model that characterizes the environmental factors affecting PP into those that control the optimal efficiency of the productivity profile and influence the relative vertical distribution of PP [19]. The advantage of the VGPM is that it incorporates satellite remote sensing data and employs minimal parameterization of input variables to derive PP [16]. Despite the understanding and knowledge of the ocean optics that determine ocean color signals and the photosynthetic process, PP derived from satellite data often have limited success in reproducing the variability observed in PP data [18,20,21]. Comparisons of PP models have shown that modeled estimates are twice as accurate as that of the carbon-based estimates [22,23]. Their application yields different results; choosing the most realistic one is therefore often a regional issue and the regional dependence of photosynthetic efficiency on hydro-optical and biochemical conditions must be taken into account [21–23]. Consequently, to obtain an appropriate match with in situ observations of PP, satellite-derived models must consider the peculiarities of regional ecosystems.

The Taiwan Strait is an important channel that transports water along the western part of Taiwan and chemical constituents between the South China Sea and East China Sea. Its alternating monsoon-forcing, complex bottom topography, and the conjunction of several current systems means that its ecosystem dynamics and biogeochemical and physical processes vary substantially in space and in time [24–26]. The warm Kuroshio current flows through the eastern part of Taiwan, and a cold dome can often be observed over the edge of the continental shelf's northeast sides [27,28]. In a review of previous studies, four major upwelling regions were identified around Taiwan, namely along the northwestern and southwestern coast of the Taiwan Strait, on the Taiwan Bank, and near the Penghu Islands [27–29]. The seasonal variation and spatial distribution of PP and phytoplankton biomass are largely controlled by the input of nutrients from various water masses [28]. The typhoon and tropical storms led to strong vertical water mixing enhanced nutrients and derived a diatom bloom around the waters of Taiwan in summer [26,30]. In particular, large-scale climatic oscillations, such as the ENSO events, also can cause sea surface temperature (SST) and PP changes on an interannual scale [28,31]. PP is an essential component of both terrestrial and aquatic ecosystems. The total fish and invertebrate production in an ecosystem-based approach to fisheries management is ultimately limited by ecosystem PP [5,6]. The high PP around the waters of Taiwan also sustains commercially important species of fish and cephalopod which develop their life cycle [28,32–34]. The development of site-specific models to estimate PP is therefore extremely desirable. Although in situ measurements of PP have been presented in previous studies (e.g., [24,26,28,35]), high-resolution PP distributions and estimations around Taiwan are rare. The objectives of this study were (1) to evaluate a regionally modified version of the VGPM using high-resolution (1.1 km$^2$) SST and Chl-a derived from satellite remote sensing data with in situ PP, and (2) to compare high-resolution VPGM PP and in situ PP for

different water masses around Taiwan to compute the root mean square difference (RMSD) and relative percentage bias (Bias). The high-resolution VPGM PP calculated from satellite data that correspond to 15 cruises provides the first view of PP around Taiwan.

## 2. Data and Methods

### 2.1. In Situ Measurements and Water Sampling

Hydrographic, optical, and biogeochemical properties were investigated in 62 sampling locations around Taiwan in ranges between 21.5°–26°N and 119°–123°E (Figure 1). These covered the period 2009 to 2013 during different seasons on the vessels of Fisheries Research I (Table 1). We defined December, January, and February as winter; March, April, and May as spring; June, July, and August as summer; and September, October, and November as autumn.

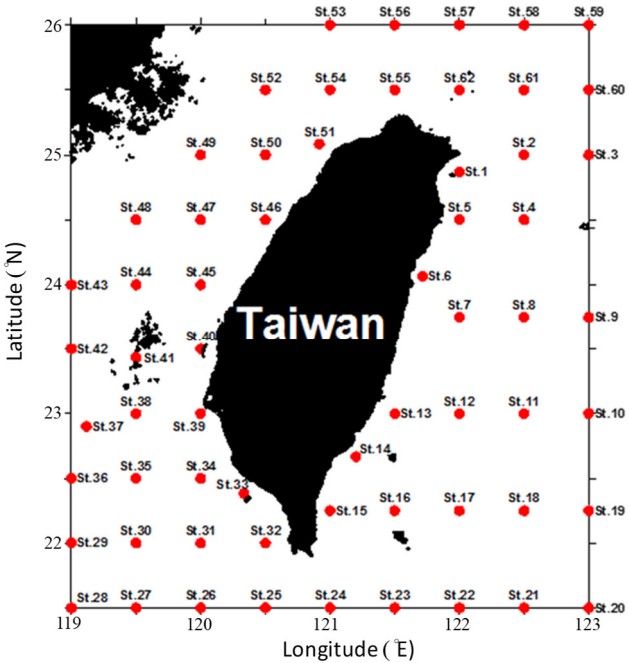

**Figure 1.** Sixty-two sampling locations of Fisheries Research I cruises around the waters of Taiwan between 21.5°–26°N and 119°–123°E.

**Table 1.** Cruise numbers, data, season, and number of PP measurement stations of Fisheries Research I from 2009 to 2013.

| Cruise No. | Date of the Cruise | Season | No. of PP Measurement Stations |
| --- | --- | --- | --- |
| FR1-2009-08-25 | 25 August–5 September, 2009 | Summer | 62 |
| FR1-2010-01-07 | 7 January–18 January, 2010 | Winter | 62 |
| FR1-2010-04-08 | 8 April–19 April, 2010 | Spring | 62 |
| FR1-2010-09-27 | 27 September–6 October, 2010 | Autumn | 62 |
| FR1-2011-01-13 | 13 January–24 January 2011 | Winter | 36 |
| FR1-2011-04-21 | 21 April–26 April, 2011 | Spring | 36 |
| FR1-2011-08-09 | 9 August–18 August, 2011 | Summer | 61 |
| FR1-2011-10-17 | 17 October–27 October, 2011 | Autumn | 62 |
| FR1-2011-12-28 | 28–31 December, 2011 1–8 January, 2012 | Winter | 59 |
| FR1-2012-04-18 | 18 April–30 April, 2012 | Spring | 62 |
| FR1-2012-08-19 | 19 August–4 September, 2012 | Summer | 55 |
| FR1-2012-11-02 | 2 November–11 November, 2012 | Autumn | 61 |
| FR1-2013-01-04 | 4 January–15 January, 2013 | Winter | 62 |
| FR1-2013-05-08 | 8 May–18 May, 2013 | Spring | 62 |
| FR1-2013-10-03 | 3 October–14 October, 2013 | Autumn | 61 |

In situ primary production was determined by light–dark bottle methods through incubation in 300 mL dissolved oxygen (DO) bottles. This technique was modified using Winkler's method. We prepared two types of DO bottles, one of which was transparent (light bottle) and the other was wrapped in aluminum foil (dark bottle). Seawater samples were collected at depths of 5, 25, and 50 m using 10 L Niskin bottles in each station, and water samples were filtered using a 300 μm filter cloth. The filtered sample was then packed into two DO bottles: one light bottle and one dark bottle. Each bottle was filled with 300 mL of seawater, following which, measurements were taken of the temperature and dissolved oxygen using a dissolved oxygen meter (YSI Model 52).

One light bottle was placed in the incubator (constant temperature of 25 °C, 4000 lux light). The dark bottle was set in the incubator, which was dark. After one day, the temperature and DO of the three bottles were measured. The PP was the difference in dissolved oxygen between the light and dark bottles, the formula for which was as follows:

In situ PP = ($[O_2]$L−$[O_2]$D) × carbon atom weight/dissolved oxygen atom weight/1 day.

$[O_2]$L: the dissolved oxygen in the light bottle after incubating for 1 day.
$[O_2]$D: the dissolved oxygen in the dark bottle after incubating for 1 day.
The PP (mg C m$^{-2}$ d$^{-1}$) was then integrated in terms of depth (m).
The depth ranged from 0–50 m.

### 2.2. Satellite-Derived PP Estimates

The PP model of the VGPM used in this study was based on Chl-a concentration, and the formulation and parameterization were recommended by Behrenfeld and Falkowski [19]. Maximum photosynthetic efficiency in VGPM is described as an optimal rate of photosynthesis ($P^B_{opt}$) in a water column normalized to Chl-a concentration (mg·C·mg·Chl$^{-1}$·h$^{-1}$). The VGPM estimates the daily integrated PP in a water column of euphotic depth ($PP_{eu}$, mg C m$^{-2}$ day$^{-1}$) as follows:

$$PP_{eu} = 0.66125 \times Chla \times P^B_{opt} \times \frac{PAR}{PAR + 4.1} \times Z_{eu} \times \mathrm{DP} \tag{1}$$

Photosynthetically available radiation (PAR) denotes daily averaged surface photosynthetic active radiation at 400–700 nm (E·m$^{-2}$·day$^{-1}$), $Z_{eu}$ denotes euphotic depth, and DP denotes a day photoperiod. $Z_{eu}$ was calculated from satellite surface chlorophyll-a concentration for lower and higher total chlorophyll conditions following Morel and Berthon [36]. $P^B_{opt}$ is expressed as a seventh-order polynomial function of SST [19], which is formulated as follows:

$$P^B_{opt} = 1.2956 + 2.749 \times 10^{-1}SST + 6.17 \times 10^{-2}SST^2 - 2.05 \times 10^{-2}SST^3 + 2.462 \times 10^{-3}SST^4 - 1.348 \times 10^{-4}SST^5 + 3.4132 \times 10^{-6}SST^6 - 3.27 \times 10^{-8}SST^7 \tag{2}$$

The satellite data used in VGPM during the study period of 2009–2013 are SST, Chl-a, and PAR. MODIS Aqua/Terra daily Level 1A were downloaded from the NASA Ocean Color website. SeaDAS v6.2 was used to process high-resolution (1.1 km) local area coverage images and Chl-a data (OC3Mv6 algorithm). Daily PAR product data were downloaded from the Ocean Productivity database. Daily SST data were extracted from NOAA AVHRR SST images and had a spatial resolution of 1.1 km. The NOAA HRPT data, including AVHRR scenes, were received at a ground station at National Ocean Taiwan University. Daily DP data were produced by the Central Weather Bureau using a Precision Spectral Pyranometer. The three categories of satellite-derived PP were calculated using different MODIS Chl-a data as follows: (1) MODIS Terra evaluated primary production (PP$_T$), (2) MODIS Aqua evaluated primary production (PP$_A$), and (3) the averaged MODIS Aqua and Terra evaluated primary production (PP$_{A\&T}$).

The divisions of waters around Taiwan also have different oceanographic characteristics. They were divided into 1° gridded areas across 36 sites (Figure 2a). Cluster analysis with normalized

Euclidean distances was used to measure levels of similarity in gridded areas in the waters around Taiwan in 2009–2013, including monthly Chl-a, SST, and PAR. Ward's method was used to illustrate the relationships between them in a dendrogram. The cluster analysis was conducted using STATISTICA 8 statistical software.

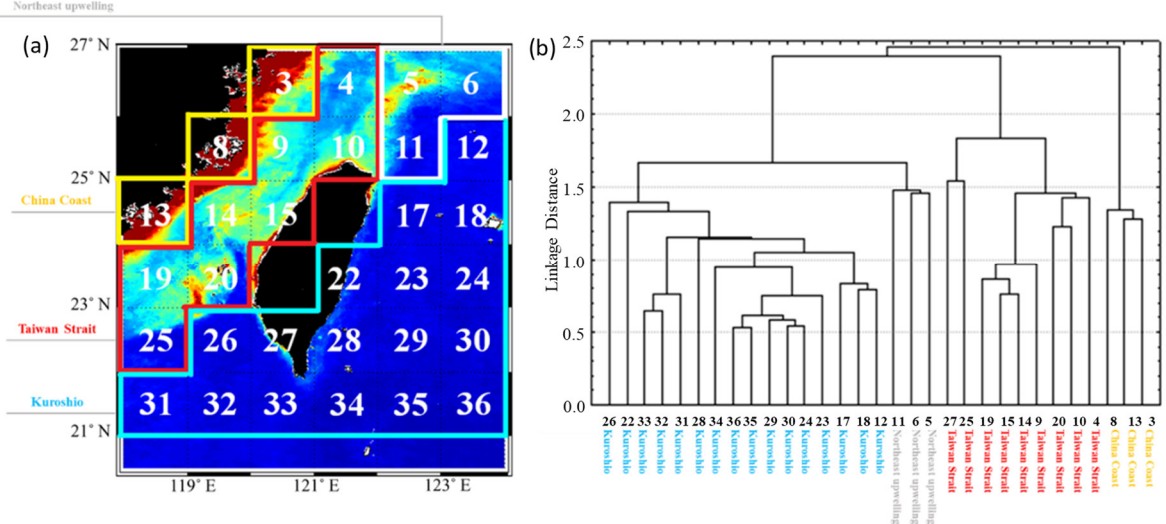

**Figure 2.** (**a**) The waters around Taiwan were divided into 1° gridded areas for 36 sites. The subareas for the China Coast water, Taiwan Strait water, Northeastern upwelling water, and Kuroshio water were divided by the cluster tree diagram results using the monthly Chl-a, SST, and PAR during study period (2009–2013) in (**b**). The monthly mean PP$_{A\&T}$ image in 2009 was used as an example in (**a**).

*2.3. Match-Up Data and the Assessment of Satellite PP Models*

To evaluate the satellite-derived PP with in situ PP, we produced pairs of collocated satellite overpasses and in situ sampling was extracted with a time difference shorter than ± 12 h. The satellite observations have spatial averages of 3 × 3 pixels around each sample site location and were compared with field measurements.

The PP values derived from the VPGM model were regressed against in situ data and a type II linear regression model was applied, as both field and modeled data are subject to errors. The slope, intercept, and the correlation coefficient (r) were then determined. For Chl-a, the regression was performed between log-transformed values. The RMSD statistic assesses model skill such that models with lower values have higher skill, and the model bias assesses whether a model over- or underestimates PP [11]. We calculated the RMSD for n samples of PP:

$$RMSD = \sqrt{\frac{\sum_{i=1}^{n} \left[ log(PP_{model,i}) - log(PP_{in\ situ,i}) \right]^2}{n}} \tag{3}$$

where $(PP_{model,i})$ modeled PP and $(PP_{in\ situ,i})$ represents in situ PP estimates at each site. To assess whether a model over- or underestimated PP, we calculated the bias of each model as follows:

$$Bias = \overline{log(PP_{model})} - \overline{log(PP_{in\ situ})} \tag{4}$$

## 3. Results

*Annual and Seasonal Trends in PP*

The spatial distribution of VPGM-based production derived from AVHRR SST and MODIS Chl-a in the waters around Taiwan showed similar seasonal spatial patterns to in situ PP (Figures 3 and 4). The highest concentration of PPs was observed along the Mainland China Coast and four major upwelling

regions (southwestern and northwestern coast, Taiwan Bank, and Penghu Islands) around the waters of Taiwan. Lower PP values were obtained across the whole year within the Kuroshio-influenced region in eastern parts. During the study periods, the total numbers of colocated satellite overpasses and in situ sampling sites were extracted with a time difference shorter than ± 12 h. This contained 102 sets for $PP_{A\&T}$, 151 for $PP_A$, and 150 for $PP_T$, respectively (Table 2).

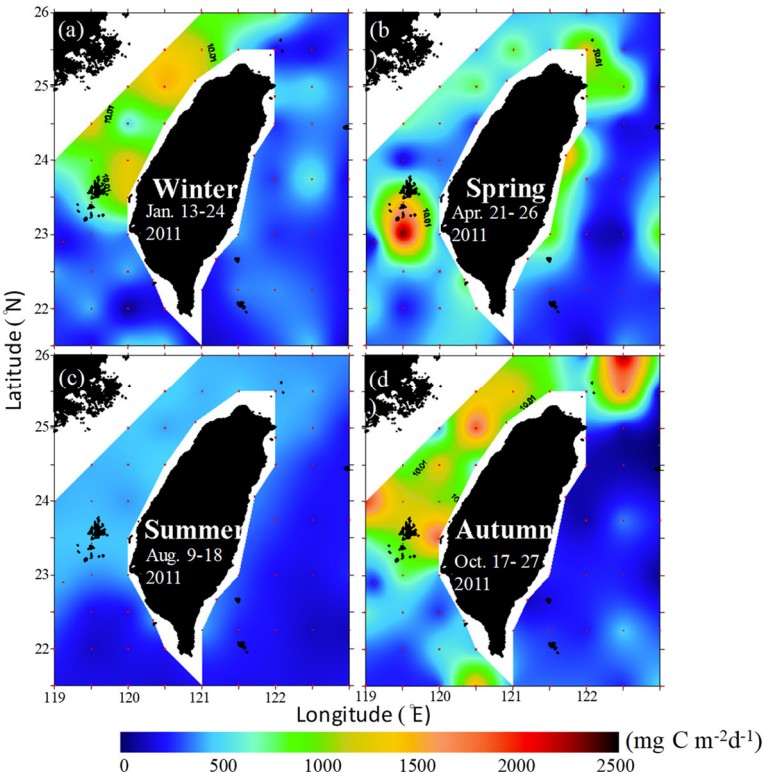

**Figure 3.** Spatial distribution of in situ PP determined using light–dark bottle methods in 2011: (**a**) winter, (**b**) spring, (**c**) summer, and (**d**) autumn.

**Table 2.** Extracted number of satellite-derived PPs with in situ PPs, correlation coefficients ($r^2$), and $p$ values for $PP_{A\&T}$, $PP_A$, and $PP_T$ for whole years and different seasons. The RMSD and bias for $PP_{A\&T}$, $PP_A$, and $PP_T$ for whole years.

| | Extracted Number | | | Correlation Coefficients | | | $p$ | | |
|---|---|---|---|---|---|---|---|---|---|
| | PP (A&T) | $PP_A$ | $PP_T$ | PP (A&T) | $PP_A$ | $PP_T$ | PP (A&T) | $PP_A$ | $PP_T$ |
| Years | 102 | 151 | 150 | 0.61 | 0.42 | 0.38 | <0.05 | <0.05 | <0.05 |
| Spring | 18 | 26 | 23 | 0.74 | 0.55 | 0.46 | <0.05 | <0.05 | <0.05 |
| Summer | 25 | 49 | 48 | 0.54 | 0.25 | 0.37 | <0.05 | <0.05 | <0.05 |
| Autumn | 52 | 55 | 59 | 0.51 | 0.46 | 0.42 | <0.05 | <0.05 | <0.05 |
| Winter | 7 | 21 | 20 | 0.33 | 0.14 | 0.07 | 0.31 | 0.22 | 0.41 |

| | RMSD | | | Bias | | |
|---|---|---|---|---|---|---|
| | PP (A&T) | $PP_A$ | $PP_T$ | PP (A&T) | $PP_A$ | $PP_T$ |
| Years | 0.37 | 0.34 | 0.34 | −0.24 | −0.197 | −0.174 |

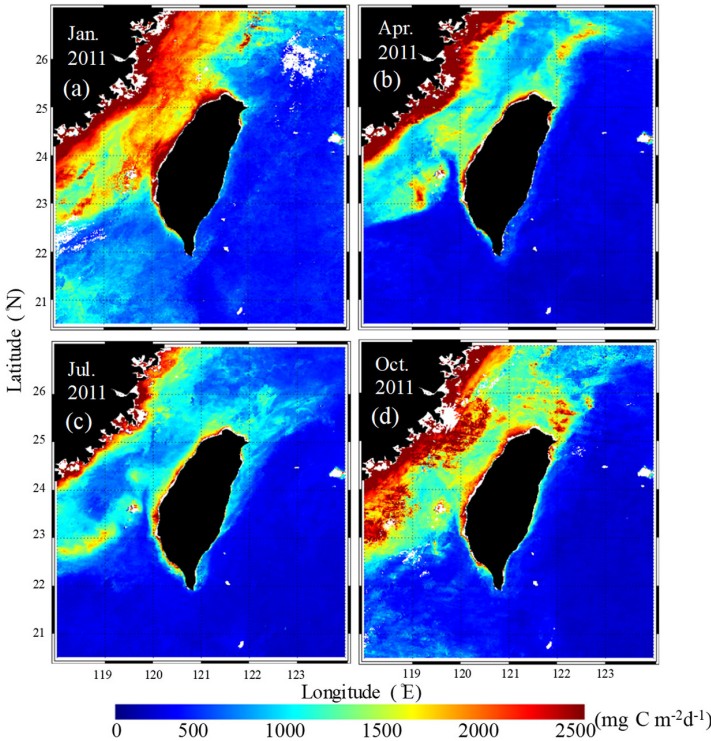

**Figure 4.** Monthly mean spatial distributions of VPGM PP derived from PP$_{A\&T}$ in 2011: (**a**) January, (**b**) April, (**c**) July, and (**d**) October.

## 4. Comparison of Satellite-Derived and in Situ PP

To validate the model results, the three satellite-derived PP values (PP$_{A\&T}$, PP$_A$, PP$_T$) were compared with in situ PP values. They were linearly correlated with the situ PP and the coefficient was higher in PPA&T (r$^2$ = 0.61) than in PP$_A$ (r$^2$ = 0.42) and PP$_T$ (r$^2$ = 0.38) throughout the year (Table 2). The highest correlations were observed in spring, especially for PP$_{A\&T}$ (r$^2$ = 0.74) (Figure 5).

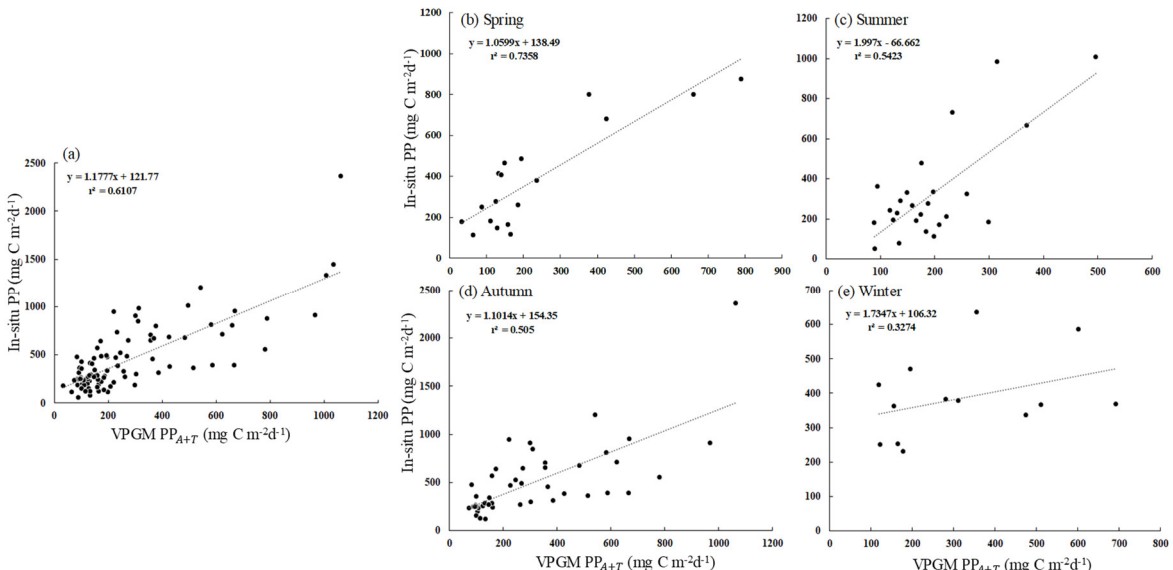

**Figure 5.** Relationship between the VPGM PP$_{A\&T}$ and in situ PP for (**a**) the whole year, (**b**) spring, (**c**) summer, (**d**) autumn, and (**e**) winter during the study period.

The lowest correlations were observed in winter and were nonsignificant. Relative to the in situ PPs, the modeled PPs ($PP_{A\&T}$, $PP_A$, $PP_T$) had a RMSD of 0.37, 0.34, and 0.34, and a bias of −0.24, −0.197 and −0.174, respectively (Table 2). The seasonal RMSR and bias were in the range of 0.03–0.09 and −0.14–0.39, respectively, for the three satellite-derived PPs. This implied that the three PPs had similar results; however, the $PP_{A\&T}$ algorithm produced slightly more accurate PP measurements than the $PP_A$ and $PP_T$ algorithm.

## 5. Cluster Analysis and Characteristics in the Subareas

The cluster analysis showed that the waters around Taiwan can be divided according to environmental conditions into four subareas: China Coast water (CCW), Taiwan Strait water (TSW), Northeastern upwelling water (NUW), and Kuroshio water (KW) (Figure 2b). In terms of the relationship between SST, MODIS Aqua Chl-a, and $PP_A$ in four subareas for each 1 °C gridded, the monthly mean SSTs were in the range of 7–30 °C in CCW, 13–30 °C in TSW, 17–25 °C in NUW, and 22–30 °C in KW, and had no clear relationships with $PP_A$ were revealed (Figure 6a). The strong correlation between $PP_A$ and MODIS Aqua Chl-a in subareas is shown in Figure 6b. The monthly mean $PP_A$ and Chl-a were in the ranges of 600–2500 mg·C·m$^{-2}$·day$^{-1}$ and 2–5 mg·m$^{-3}$ in CCW, 500–1500 mg C m$^{-2}$·day$^{-1}$ and 1–3 mg·m$^{-3}$ in TSW, 500–1000 mg C m$^{-2}$·day$^{-1}$ and 1–2 mg·m$^{-3}$ in NUW, and 0–500 mg C m$^{-2}$·day$^{-1}$ and 0–1.5 mg·m$^{-3}$ in KW.

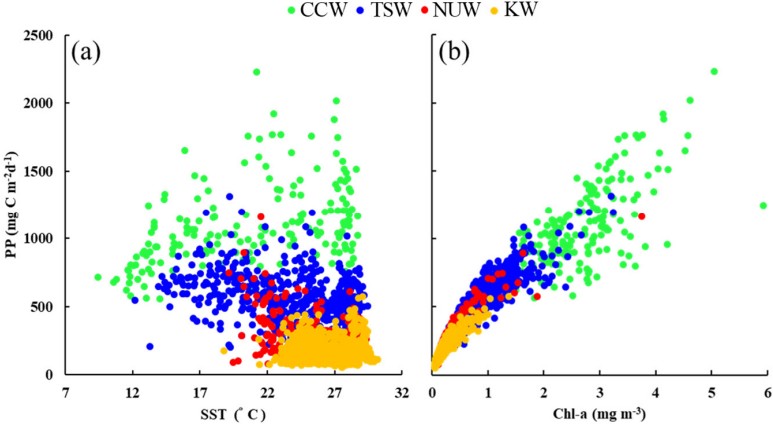

**Figure 6.** Relationship between the monthly mean of (**a**) AVHRR SST and VPGM $PP_A$, and (**b**) MODIS Aqua Chl-a and VPGM $PP_A$ for each 1 °C gridded area in the four subareas of CCW (green circles), TSW (blue circles), NUW (red circles), and KW (yellow circles) from 2009 to 2013.

The comparison of in situ data and model data for each subarea is presented in Table 3. The $PP_A$ had significant correlations with in situ PP in the TSW ($r^2 = 0.26$), NUW ($r^2 = 0.37$) and KW ($r^2 = 0.14$). $PP_{A+T}$ only had significant correlations with in situ PP in the TS, and $PP_T$ had no significant correlations in any of the subareas. The RMSD values were higher in the KW, ranging between 0.06–0.87, with an almost negative bias in the range of −0.74 to 0.38 (Figure 7). The RMSD in the TSW, NUW, and CCW were in the range of 0.07–0.67 with bias in the range of −0.49 to 0.27.

**Table 3.** Extracted number (n) of satellite-derived PPs with in situ PPs, correlation coefficients ($r^2$), and *p* values for $PP_{A\&T}$, $PP_A$, and $PP_T$ for whole years in four subareas.

|  | China Coast | | | Taiwan Strait | | | Northeast Upwelling | | | Kuroshio | | |
|---|---|---|---|---|---|---|---|---|---|---|---|---|
|  | *n* | $r^2$ | *p* | *n* | $r^2$ | *p* | *n* | $r^2$ | *p* | *n* | $r^2$ | *p* |
| $PP_{A\&T}$ | 3 | 0.16 | 0.51 | 39 | 0.08 | <0.05 | 12 | 0.01 | 0.79 | 48 | 0.02 | 0.07 |
| $PP_A$ | 4 | 0.33 | 0.31 | 52 | 0.26 | <0.05 | 12 | 0.37 | <0.05 | 83 | 0.14 | <0.05 |
| $PP_T$ | 3 | 0.44 | 0.54 | 50 | 0.04 | 0.09 | 14 | 0.01 | 0.7 | 83 | 0.02 | 0.13 |

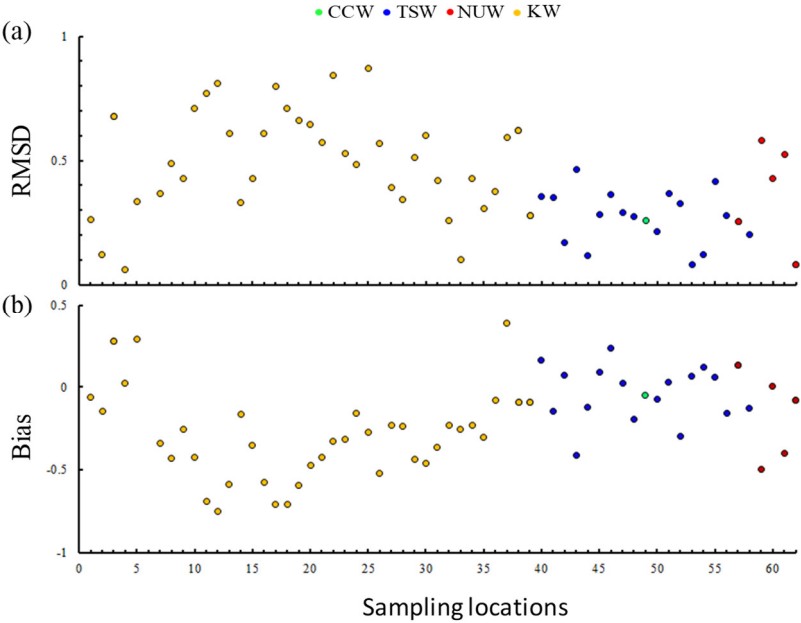

**Figure 7.** (**a**) RMSD and (**b**) bias for the 62 sampling locations in the four subareas of the CCW (green circles), TSW (blue circles), NUW (red circles), and KW (yellow circles).

## 6. Discussion

Primary producers reside at the base of food webs, and thus, drive ecosystem dynamics through bottom-up forcing [11]. The global biogeochemical cycles of major elements, particularly the carbon cycle, are greatly influenced by primary producers [2]. The primary producers convert the inorganic to organic carbon by photosynthesis in the light environment, and the carbon production is referred to as gross PP [37]. Net PP is gross PP minus the phytoplankton's respiration, and it supplies to all heterotrophs in the oceans. Net community production is gross PP minus respired by autotrophs and heterotrophs [37]. Both net PP and net community production play an important factor for biological carbon circulation. Therefore, understanding the spatial and temporal dynamics of PP is invaluable in earth and life science research [38,39].

The present study provided the first assessment of a satellite-derived VPGM PP model with in situ PP estimates in the regional waters around Taiwan. In light–dark oxygen methods, not only phytoplankton but also heterotrophic bacteria and zooplankton are in the bottles. The charge in dissolved oxygen of the light bottle is affected by photosynthesis and community respiration, and the gross PP can be estimated by net community production and community respiration [37]. The light–dark oxygen method used in the present study and the $^{14}$C method are often used to measure in situ marine PP. The light–dark oxygen method was the main approach to measure PP before the $^{14}$C method was invented [1], the latter of which is more sensitive and offers good precision, although a radioisotope needs to be added to the water samples. The acquisition, use, and disposal of the radioisotope requires specific procedures and incurs high costs. Nevertheless, it yields reliable data through a careful process using oxygen electrodes [1].

In situ measurements of PP are spatially and temporally limited and require multiple integrated sampling approaches. The use of ocean color data in PP models provides an attractive alternative to field estimations as it offers an estimation at a high spatial and temporal resolution. AVHRR SST and MODIS Chl-a have long been used to study marine characteristics; however, their availability is seriously reduced by cloud coverage. The coverage provided by AVHRR SST and MODIS Chl-a daily images in the current study were in the range 20%–80% (Figure 8 as examples), and notably lower in wintertime (<30%). Although the availability of data from microwave observations was almost 100% [40], the disadvantage of a low spatial resolution meant that it was not appropriate for use in

coastal areas. The limited match-up data between the satellite-derived data and in situ measurements were relaxed to within ± 12 h in the present study, and this time difference may have affected the matching accuracy. For example, Lee et al. [41] compared the MODIS SSTs with in situ SSTs and suggested that if the time difference was ± 3 h, this would produce the smallest bias but with a lower match-up of usable data.

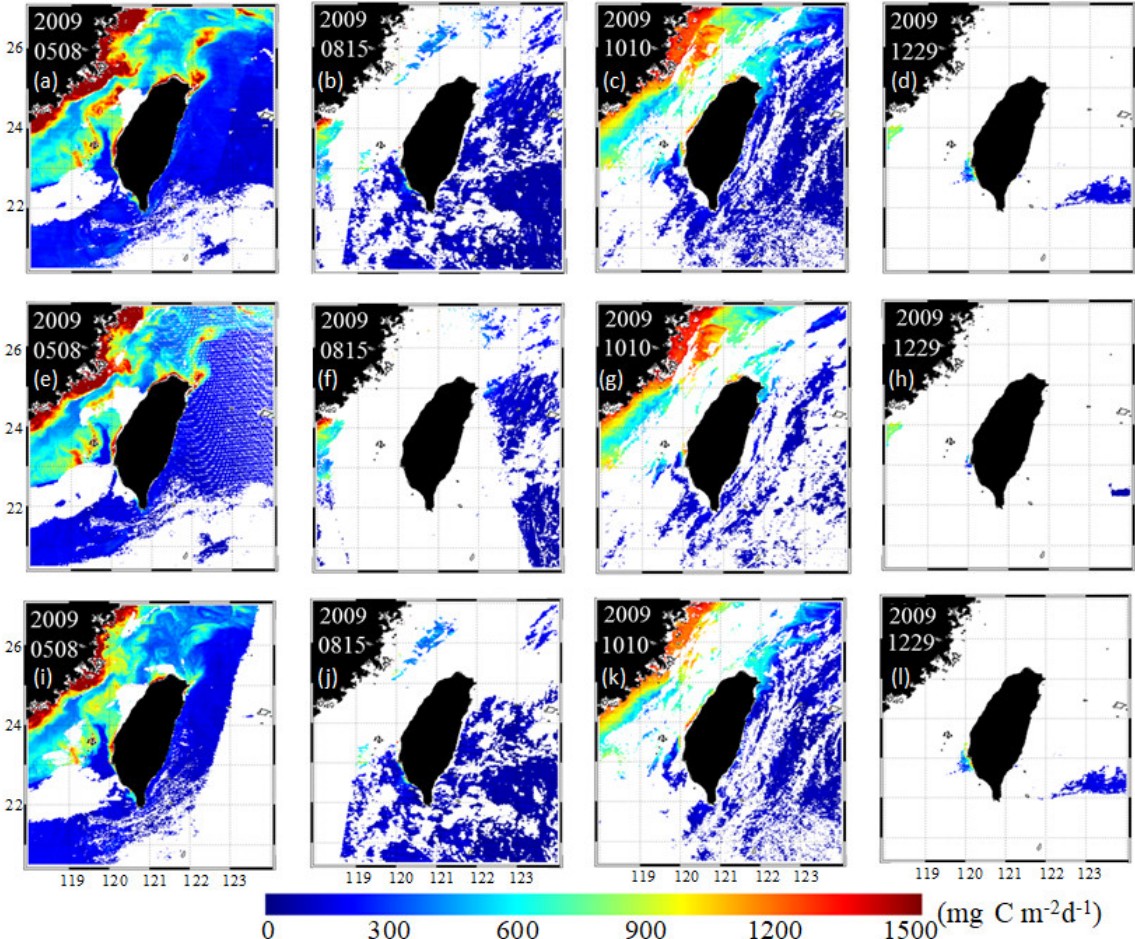

**Figure 8.** Daily spatial distribution maps of (**a–d**) PP$_{A\&T}$, (**e–h**) PP$_A$, and (**e–h**) PP$_T$ in May, August, October, and December 2009.

To make best use of this technique, it is essential to assess the accuracy of the satellite-derived products to determine the uncertainty of the data input into the models [14]. The bias and RMSD between AVHRR SST and in situ data were 0.01 and 0.64 °C and high accuracy than the MODSI SST (bias = 0.03 °C, RMSD = 0.75 °C) in the waters around Taiwan [41,42]. For the MODIS Chl-a data, the previous study suggested that in coastal or upwelled waters, the blue region of the water that left a radiance signal used in standard Chl-a satellite algorithms was affected by colored dissolved organic matter and detrital material in addition to phytoplankton. This resulted in the decreasing accuracy of Chl-a and therefore PP [18,20]. The matching accuracy OC3M algorithm was within 11% of in situ Chl-a in the Arabian Sea and performed better than the Garver Siegel Maritorena Model and Generalized Inherent Optical Property Chla algorithms [43]. The satellite Chl-a estimates tended to be larger than in situ reference values, and also revealed that a nonuniform Chl-a distribution in the water column can be a factor alongside the documented overestimation tendency when larger optical depth match-up stations are considered [44].

When the Chl-a derived from MODIS Aqua and Terra is compared, the MODIS Terra is more accurate in the coastal waters of the Arabian Sea [45], possibly due to differences in sensor design and

time differences between the satellites' overpasses. Applying standard products of satellite-derived PP for research in the open ocean is acceptable; however, using such products in regional studies remains questionable. This is especially so when the shelf sea areas are dominated by large rivers containing a large amount of suspended particles and color-dissolved organic matter [26,35]. The estimation errors of satellite-derived PP were often the result of incorrectly applying ocean color chlorophyll algorithms and inaccurate PP data [46]. In our case, the highest correlation coefficients were obtained in the $PP_{A+T}$ around the waters of Taiwan in the spring. However, the highest significant correlation coefficients in the four subareas were observed for $PP_A$. The lowest correlations for $PP_{A+T}$, $PP_T$, and $PP_A$ all occurred in wintertime. Although the RMSD and bias were lowest in $PP_A$, the difference was not significant. We found that VPGM estimated PPs were always underestimated with satellite-derived PPs, which may be due to errors in the $P^B_{opt}$ calculated as a function of SST [47]. However, this photosynthetic parameter also strongly depends on factors such as nutrient supply, irradiance, and dominating phytoplankton species.

Satellite SST represents temperature only in the uppermost ocean layer and $P^B_{opt}$, being a function of SST, differs in the lower layers, providing different photosynthetic efficiency [21]. The VGPM is one of the most widely known and applied depth-integrated/wavelength-integrated models. However, it has rarely been applied to coastal waters. Lobanova et al. [21] compared the accuracy of PP derived from the VPGM, the Platt and Sathyendranth model, and the Absorption-Based Model with in situ data in the North East Atlantic Ocean. The results revealed that the Platt and Sathyendranth model and VPGM had similar accuracy, whereas the Absorption-Based Model was not suitable for the study region. Although using in situ $P^B_{opt}$ and $Z_{eu}$ may have significantly improved the estimation of VPGM PP [16], the scales of in situ measurements were too short to provide adequate coverage of high-quality regional temporal and spatial variations. Improvements in the accuracy of Chl-a from other ocean color sensors, including Medium Spectral Resolution Imaging Spectrometer and Ocean-Colour Climate Change Initiative data, will ultimately lead to an improvement in satellite PP algorithms for further research.

The VPGM PP in the four subareas of CCW, TSW, NUW, and KW also displayed similar features to Chl-a variations with the highest PP in the CCW and lowest in the KW. The RMSD was higher in the KW with an almost negative bias. The $PP_A$ had significant correlations with in situ PP in the TS, NUW, and KW; however, the sampling locations were insufficient to provide significant results in the CCW. The availability of light, the source of energy for photosynthesis; mineral nutrients, (the building blocks for new growth); and temperature, which affects metabolic rates, play crucial roles in regulating PP in the ocean [48,49]. The dominant primary producers in the Taiwan Strait are nano- and pico-phytoplankton [28]. The contribution of the microbial food web to the traditional food web is estimated to be approximately 30%, implying it has fundamental significance in the Taiwan Strait [28]. The high salinity and temperature with low nutrients originate from the TSW source in the South China Sea and KW in summertime; the strong northeastern winds then push the fresh, cold, nutrient-rich CCW southward along the western part of the Taiwan Strait [28,50]. In addition, the KW flowing through the eastern part of Taiwan is relatively deficient in nutrients, and the PP is also lower than in the other currents around Taiwan [51]. However, the nutrients increase from a depth of 200 m under short-term climatic variations such as typhoons [52], and may cause higher RMSD and bias in KW. The spatial distribution and seasonal variation of phytoplankton biomass and primary productivity are largely controlled by the input of nutrients from various water masses.

## 7. Conclusions and Future Research

The present study provided the first assessment of a satellite-derived VPGM PP model with in situ PP estimates in the regional waters around Taiwan. Understanding the PP of waters affected by global warming is critical. In particular, the East China Sea is among the large marine ecosystems that are warming most rapidly. Furthermore, from the 1950s to 2000s, the increased SST has caused 20 °C isotherms in the Taiwan Strait to gradually shift northward in the winter [53]. Climate change caused by global warming along with changes in water temperature also affect the productivity, catchability,

and fishing pressure of fish species. It is important to quantify and understand the sources of variation in marine PP and increase confidence in the predictions of future fisheries yielded under uncertainty over future PP and its transfer to higher trophic levels [54]. Our results revealed that the VGPM PP derived from AVHRR SST and three types of MODIS Chl-a were linearly correlated with in situ PPs, as determined by light–dark bottles. The correlation coefficients were highest in the $PP_{A\&T}$ around the waters of Taiwan, especially in springtime. However, the highest significant correlation coefficients in the four subareas were observed in $PP_A$ and wintertime.

Satellite models underestimate in situ PP, probably due to the depth of the phytoplankton in the water column, short-term climatic variations, and optically complex shelf waters. To better understand the PP around Taiwan with complex current systems, substantially more measurements are required across multiple years and seasons. Although initially, these can be used to quantify the productivity of different water masses, they are eventually required to further validate the available biogeochemical models in order to scale up relatively sparse measurements through time and space [18]. Additional studies include making a further comparison of in situ marine PP with the $^{14}$C method and multi satellite-derived data, such as Ocean-Colour Climate Change Initiative (OC-CCI) data obtained from merged information derived from ocean color sensors. The VPGM is a commonly used model and exhibited significant correlations with in situ data in the present study. However, the simplest model formulation of the Eppley-Square-Root model [15] was tested as the highest skill and lowest bias model in the western boundary of East Australia [18]. The other VPGM-based models, the VGPM-Eppley model [2], and VGPM-Kameda model [17], will provide the crucial next step in conducting more comprehensive investigations by re-parameterizing the original relationships in accordance with in situ data.

**Author Contributions:** K.-W.L. led the study design and wrote the article. L.-J.L. and P.-Y.H. contributed materials of remote sensing data and analysis models. C.-H.L. and S.-Y.C. contributed in situ measurements and water sampling. All authors have read and agreed to the published version of the manuscript.

**Funding:** This research received no external funding.

**Acknowledgments:** This study was financially supported by the Council of Agriculture (108AS-9.2.1-FA-F1, 108AS-25.2.1-AI-A2, 109AS-9.2.2-FA-F1(2) and 109AS-20.2.1-AI-A2) and the National Science Council (MOST 107-2611-M-019-017 and MOST 108-2611-M-019-007).

**Conflicts of Interest:** The authors declare no conflict of interest

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
