# Peer review of "Validation of a Primary Production Algorithm of Vertically Generalized Production Model Derived from Multi-Satellite Data around the Waters of Taiwan"

_remotesensing, doi:10.3390/rs12101627_

Round 1
Reviewer 1 Report
Comments to the article:
Validation of a primary production algorithm of vertically generalized production model derived from multi-satellite data around the waters of Taiwan.
Comments generals.
I consider that the authors did a good job in order to develop a model with a greater approximation for the regional analysis of PP. However, there is a lack of information in the introduction about the importance of these types of models and their scientific application in oceanographic and biological aspects, which will help to understand the relationship between the resource environment and to explain changes in ecosystems and facilitate the use of these models in said areas. There are also methodological shortcomings regarding the integration of statistical analyzes which will give strength to the results obtained. Therefore, I recommend a major revision of the article for publication in Remote Sensing.
Introduction.
Line 43-44. Indicate what are those short scales… Add reference.
Line 44-46. Explain why?
Since in previous lines they mention that the primary production is reflected in short time scales. Furthermore, the analysis of Chl a as an indicator of primary production in coastal regions can be very useful to know the health status of coastal ecosystems on relatively short scales (for example: weeks and months).
Line 56-68. Considering the limitations of bio-optical models, regarding the integration of specific characteristics of coastal ecosystems, would it not be better to continue using direct satellite data? If not, explain better why use VGPM.
Line 69-74. For the understanding of the oceanography of a study region it is not enough to put general characteristics. Therefore, the authors must describe these characteristics in the region as: ocean-atmosphere interactions, which structures at the bottom of the ocean and which mesoscale phenomena in the water column influence bio-geochemical processes, which will have an effect on the concentration of Chl a.
The objective of developing a regional PP model must be accompanied by regional studies of the importance of the use of these models in terms of their use to determine the health of ecosystems, population dynamics of species of biological importance, fishing, aquaculture, climate change , etc. I recommend the integration of this literature and give greater importance to the proposed regional model.
Methodology:
Line 139-144. Regionalization, a statistical analysis of the monthly PPP and T data must be integrated in order to check if there is a statistically significant difference between the regions for all the years of the groups observed in the cluster analysis. This cluster analysis should be verified since it is not correct to only take the year 2009 instead of the entire time series (2009-2013), since the concentration variability of Chl a and SST is not the same every year. Therefore, the groups in the cluster analysis may change or indeed may remain the same. This will be verified by the table of the statistical analysis of the data with the variability of Chla and SST. Include Average, SD, and sub-index that denotes significant differences between regions.
Do the same analysis for the data in situ and check to see if the same behavior exists in terms of differences between regions.
Line 197-206. Integrate the statistical analysis of the data to corroborate these results. Since without the statistics it cannot be said that there are differences between the regions.
Discussion.
To the extent that suggestions and pertinent corrections are made in the article, it will be reflected in the results and discussion, as well as in the conclusions.
Author Response
Comments generals.
I consider that the authors did a good job in order to develop a model with a greater approximation for the regional analysis of PP. However, there is a lack of information in the introduction about the importance of these types of models and their scientific application in oceanographic and biological aspects, which will help to understand the relationship between the resource environment and to explain changes in ecosystems and facilitate the use of these models in said areas. There are also methodological shortcomings regarding the integration of statistical analyzes which will give strength to the results obtained. Therefore, I recommend a major revision of the article for publication in Remote Sensing.
=> Thank you for your important comments and encourage. We had added more information about the importance of the VGPM models and their scientific application in oceanographic and biological aspects in the “Introduction”. For the statistical analyzes, there may had some typo in the sentences and figure captions, and led to misunderstanding the methods and time periods we used in the manuscript. We had revised “Data and Methods” more clear and gave more explanation in the replay below. We hope that the manuscript after revision can fulfill the requirement from the referees.
Introduction.
(1) Line 43-44. Indicate what are those short scales… Add reference.
=> We had revised the sentence as below: These shipboard measurements of the snapshot sections vary over short temporal and spatial scales [7–10].
(2) Line 44-46. Explain why? Since in previous lines they mention that the primary production is reflected in short time scales. Furthermore, the analysis of Chl a as an indicator of primary production in coastal regions can be very useful to know the health status of coastal ecosystems on relatively short scales (for example: weeks and months).
=> In the sentence, we hope to mention the in situ shipboard measurements had the limitations because of the short temporal and spatial scales of the snapshot sections, and need to rely on remote sensing data and models. Thus, we had revised the sentence as below: Scaling these relatively separate in situ measurements of the snapshot sections to a regional scale, let alone basin or global scale projections, therefore remains a significant challenge and need to rely on remote sensing data and models [11–14].
(3) Line 56-68. Considering the limitations of bio-optical models, regarding the integration of specific characteristics of coastal ecosystems, would it not be better to continue using direct satellite data? If not, explain better why use VGPM.
=> Thank you for your comments. We added the sentence to introduce the limitations for using direct satellite data as specific characteristics of PP as below: Remote sensing of ocean color cannot provide adequate information on oceanic PP without the support of models and sea truth data [15, 19].
(4) Line 69-74. For the understanding of the oceanography of a study region it is not enough to put general characteristics. Therefore, the authors must describe these characteristics in the region as: ocean-atmosphere interactions, which structures at the bottom of the ocean and which mesoscale phenomena in the water column influence bio-geochemical processes, which will have an effect on the concentration of Chl a.
=> We added more sentences about the effect of the ocean-atmosphere interactions on the concentration of PP as below: The typhoon and tropical storms led strong vertical water mixing enhanced nutrient and derived a diatom bloom around the waters of Taiwan in summer [26, 30]. In particular, large-scale climatic oscillations such as the ENSO events also can cause sea surface temperature (SST) and PP changes on an inter-annual scale [28, 31].
New references:
- Hung, C.C.; Chung, C.C.; Gong, G.C.; Jan, S.; Tsai, Y. et al. Nutrient supply in the southern East China Sea after typhoon Morakot. J. Mar. Res.2013, 71(1–2), 133–149.
31 Tzeng, M.T.; Lan K.W.; Chan, J.W. Interannual Variability of Wintertime Sea Surface Temperatures in the Eastern Taiwan Strait. J. Mar. Sci. Technol.-Taiwan 2012, 20(6), 702–712.
(5) The objective of developing a regional PP model must be accompanied by regional studies of the importance of the use of these models in terms of their use to determine the health of ecosystems, population dynamics of species of biological importance, fishing, aquaculture, climate change , etc. I recommend the integration of this literature and give greater importance to the proposed regional model.
=> Thank you for important comment, and we added more sentences to give greater importance to the proposed regional model as below: give greater importance to the proposed regional model: The total fish and invertebrate production in an ecosystem-based approach to fisheries management are ultimately limited by ecosystem PP [5, 6]. The high PP around the waters of Taiwan also sustains commercially important species of fish and cephalopod which develop their life cycle [28, 32–34].
New references:
- Lee, K.T.; Liao, C.H.; Su, W.C.; Hsieh, S.H.; Lu, H. J. The fishing ground formation of sergestid shrimp (Sergia lucens) in the coastal waters of southwestern Taiwan. J. Mar. Sci. Technol.-Taiwan2004, 12(4), 265–272.
- Lu, H.J.; Lee, H.L. Changes in the fish species composition in the coastal zones of the Kuroshio Current and China Coastal Current during periods of climate change: Observations from the set-net fishery (1993–2011). Fish Res.2014, 155, 103–113.
34 . Liao, C.H.; Lan K.W.; Ho H.Y.; Wang K.Y.; Wu, Y.L. Variation in the catch rate and distribution of swordtip squid (Uroteuthis edulis) associated with factors of the oceanic environment in the southern East China. Mar. Coast. Fish. 2018, 10, 452–464.
(6) Methodology:
Line 139-144. Regionalization, a statistical analysis of the monthly PP and T data must be integrated in order to check if there is a statistically significant difference between the regions for all the years of the groups observed in the cluster analysis. This cluster analysis should be verified since it is not correct to only take the year 2009 instead of the entire time series (2009-2013), since the concentration variability of Chl a and SST is not the same every year. Therefore, the groups in the cluster analysis may change or indeed may remain the same. This will be verified by the table of the statistical analysis of the data with the variability of Chla and SST. Include Average, SD, and sub-index that denotes significant differences between regions.
Do the same analysis for the data in situ and check to see if the same behavior exists in terms of differences between regions.
Line 197-206. Integrate the statistical analysis of the data to corroborate these results. Since without the statistics it cannot be said that there are differences between the regions.
=> We agreed for your comments, and we also process our analysis by using entire time series, not only 2009 in our original manuscript. There may had some typo in the sentences and figure captions, and led to misunderstanding the methods and time periods we used in the manuscript. We used the entire time series (2009-2013) to process our analysis including cluster analysis, RMSD and bias, not only 2009. I think the problem is the figure caption of Figure 2. Our original intention is to refer the background PP images in Figure 2(a) used the 2009 image as an example, and may easily lead to misunderstanding by the reader. We had revised the figure captions of Figure 2 as below: Figure 2. (a) The waters around Taiwan were divided into 1° gridded areas for 36 sites. The subareas for China Coast water, Taiwan Strait water, Northeastern upwelling water, and Kuroshio water were divided by the cluster tree diagram results using the monthly Chl-a, SST, and PAR during study period (2009–2013) in Figure 2 (b). The monthly mean PPA&T image in 2009 was used as an example in (a). We also added more description about our study period in the “Satellite-derived PP estimates” as below: The satellite data used in VGPM during the study period of 2009–2013 are SST, Chl-a, and PAR. Cluster analysis with normalized Euclidean distances was used to measure levels of similarity in gridded areas in the waters around Taiwan in 2009–2013, including monthly Chl-a, SST, and PAR.
Discussion.
To the extent that suggestions and pertinent corrections are made in the article, it will be reflected in the results and discussion, as well as in the conclusions.
=> As we mentioned above, we had revised the typo in the sentences and figure captions in “Satellite-derived PP estimates”, and hope the reader can reduce misunderstandings about our analytical methods and study period we used in the manuscript.
The authors greatly appreciate helpful comments of the Reviewer.
Reviewer 2 Report
It is a pleasure for a reviewer to find a contribution that is nearly perfect in its conception. organization, layout, and significance of conclusions. I enjoyed reading this paper on primary production around the island of Taiwan. The thesis of the project is to compare results of in situ analysis of primary prodution in surface waters (based on 62 sample sites completely surrounding the island) with satellite reconnaissance looking at the same feature through four annual seasons.
Figures 3 and 4 make a convincing comparison between the two methods, which show strong overlap in results. The satellite images clearly show much more detail than apparent from the 62 dark/white bottle sample localities.
The paper reads smoothly and the graphics are excellent. The only flaw I noticed was in Table 1, where some of the columns are out of sync.
I have been a party to similar studies on the west coast of N. America, but in the Taiwan study I was much struck by the argument that satellite surveys need to continue on a regular basis to check on changes due to global warming, especially in the East China Sea.
Author Response
It is a pleasure for a reviewer to find a contribution that is nearly perfect in its conception. organization, layout, and significance of conclusions. I enjoyed reading this paper on primary production around the island of Taiwan. The thesis of the project is to compare results of in situ analysis of primary production in surface waters (based on 62 sample sites completely surrounding the island) with satellite reconnaissance looking at the same feature through four annual seasons.
Figures 3 and 4 make a convincing comparison between the two methods, which show strong overlap in results. The satellite images clearly show much more detail than apparent from the 62 dark/white bottle sample localities.
The paper reads smoothly and the graphics are excellent. The only flaw I noticed was in Table 1, where some of the columns are out of sync.
I have been a party to similar studies on the west coast of N. America, but in the Taiwan study I was much struck by the argument that satellite surveys need to continue on a regular basis to check on changes due to global warming, especially in the East China Sea.
=> We’re so appreciated for your positive comments, and we also had revised the columns align in Figure 1. Although our results revealed not all the subareas and seasons had high correlations between VPGM PPs and in situ PPs, but the present study provided the satellite-derived PP model was useful to view the characteristics of PPs around the waters of Taiwan. The long-term fixed in site and satellite surveys are very important and necessary to examine the marine environment changes caused by the global warming.
The authors greatly appreciate helpful comments of the Reviewer.
Reviewer 3 Report
The manuscript presents a study where in-situ PP is compared to satellite estimations in the waters around Taiwan. This is a nicely performed study with valuable results that will be useful for the larger larger community. I have a couple of question and comments:
1. The authors follow general practice by comparing absolute values of in-situ and satellite PP, but I don’t think that is a very useful approach. I would prefer to the same analysis being conducted on growth rates or at least Chl-normalized primary productions. The reason for this request is that the current comparison mainly show that areas with very high Chl concentrations also have very high primary production, which is to be expected. The ambition of VGPM is to estimate growth rates, I.e. how fast the community doubles per day. This is a much more interesting validation metric of the product. I'm not criticizing the authors at all since they follow standard precise, I just think that a growth rate comparison would be generally more useful.
2. Why do you use AVHRR and not MODIS SST? Is this a skin or bulk measurement of SST?
3. I would like more detailed discussion about that kind of Primary production that the in-situ method measures - Gross Primary Production, Net Primary Production, or Net Community Production?
4. Aren't regression correlations normally presented as Coefficients of Determination (R^2), not Correlations Coefficient (R)?
5. The paragraph on line 250-261 could be complemented with a discussion about the presence of Class I and class II waters.
6. It sounds from the introduction that one ambition was to re-parametrize VGPM for regional conditions was this not feasible?
Author Response
The manuscript presents a study where in-situ PP is compared to satellite estimations in the waters around Taiwan. This is a nicely performed study with valuable results that will be useful for the larger community. I have a couple of question and comments:
(1) The authors follow general practice by comparing absolute values of in-situ and satellite PP, but I don’t think that is a very useful approach. I would prefer to the same analysis being conducted on growth rates or at least Chl-normalized primary productions. The reason for this request is that the current comparison mainly show that areas with very high Chl concentrations also have very high primary production, which is to be expected. The ambition of VGPM is to estimate growth rates, I.e. how fast the community doubles per day. This is a much more interesting validation metric of the product. I'm not criticizing the authors at all since they follow standard precise, I just think that a growth rate comparison would be generally more useful.
=> Thank you for your important comment, and we agree that you comment could be our future work to analysis the growth rates of primary productions. We also added some reference to mention the important of the primary production model and can’t only use Chl-a as an primary production index as below: Remote sensing of ocean color cannot provide adequate information on oceanic PP without the support of models and sea truth data [15, 19].
(2) Why do you use AVHRR and not MODIS SST? Is this a skin or bulk measurement of SST?
=> Because compared with the bias and RMSD between AVHRR SST, MODSI SST and in situ data in waters around Taiwan, the AVHRR SST (bias: 0.01°C; RMSE: 0.64°C, Lee et al., 2005) had lower bias and RMSE than MODIS SST (bias: 0.03°C; RMSE: 0.75°C, Lee et al.,2010). We also added the comparison of bias and RMSD of AVHRR and MODSI SST in the “Discussion” as below: The bias and RMSD between AVHRR SST and in situ data were 0.01°C and 0.64 °C and high accuracy than the MODSI SST (bias=0.03 °C, RMSD=0.75 °C) in waters around Taiwan [43, 44].
New reference:
- Lee, M.A.; Tzeng, M.T.; Hosoda, K.; Sakaida, F.; Kawamura, H. et al. Validation of JAXA/MODIS sea surface temperature in water around Taiwan using the Terra and Aqua satellites. Terr. Atmos. Ocean. Sci.2010, 21(4), 7.
(3) I would like more detailed discussion about that kind of Primary production that the in-situ method measures - Gross Primary Production, Net Primary Production, or Net Community Production?
=> We had added more discussion about the Gross Primary Production, Net Primary Production, or Net Community Production as below:1. The primary producers convert the inorganic to organic carbon by photosynthesis in light environment, and the carbon production is referred to as gross PP [37]. Net PP is gross PP minus the phytoplankton’s respiration, and it supplies to all heterotrophs in the oceans. Net community production is gross PP minus respired by autotrophs and heterotrophs [37]. Both net PP and Net community production play an important factor for biological carbon circulation. 2. In light–dark oxygen methods, not only phytoplankton but also heterotrophic bacteria and zooplankton are in the bottles. The charge in dissolve oxygen of the light bottle is effect by photosynthesis and community respiration, and the gross PP can be estimated by net community production and community respiration [37].
New reference:
- Howarth, R.W.; Michaels, A.F. Light and dark bottle oxygen technique. In: Methods in Ecosystem Science edited by Sala O.E.; Jackson, R.B.;Moone H.A.; Howarth, R.W. 2000, Springer-Verlag, New York, U.S.A. 74–80
(4) Aren't regression correlations normally presented as Coefficients of Determination (R^2), not Correlations Coefficient (R)?
=>Corrected. We had revised the r to r2 through the manuscript.
(5) The paragraph on line 250-261 could be complemented with a discussion about the presence of Class I and class II waters.
=> Thank you for comments. However, we had given the discussion about the oceanography and primary producer characteristic in the four subareas of CCW, TSW, NUW, and KW. To do the further analysis about the different class waters and give more discussion may suitable for our crucial next step.
(6) It sounds from the introduction that one ambition was to re-parametrize VGPM for regional conditions was this not feasible?
=> To re-parametrize VGPM for regional conditions is feasible. However, as we mentioned in the “Conclusions and future research” that we need to collect more in site marine PP with the 14C method and multi satellite-derived data first, and compared other VPGM based models. It will provide more comprehensive investigations by re-parameterizing the original relationships in accordance with in situ data.
The authors greatly appreciate helpful comments of the Reviewer.
Round 2
Reviewer 1 Report
Dear Authors, I consider that your corrections are sufficient to give greater importance to your manuscript and that it can be published in Remote Sensing.
All research work regarding the proposal of models and physical and biological data must undoubtedly have the application and importance in the effects of the ecosystem as well as previous antecedents that can be improved. For this reason I accept your work for publication.
Reviewer 3 Report
Looks good to me! A couple of minor typos here and there.